# ON REPRESENTING (ANTI)SYMMETRIC FUNCTIONS

## ABSTRACT

Permutation-invariant, -equivariant, and -covariant functions and anti-symmetric functions are important in quantum physics, computer vision, and other disciplines. (Anti)symmetric neural networks have recently been developed and applied with great success. A few theoretical approximation results have been proven, but many questions are still open, especially for particles in more than one dimension and the anti-symmetric case, which this work focusses on. More concretely, we derive natural polynomial approximations in the symmetric case, and approximations based on a *single* generalized Slater determinant in the anti-symmetric case. Unlike some previous super-exponential and discontinuous approximations, these seem a more promising basis for future tighter bounds. In the supplementary we also provide a complete and explicit universality proof of the Equivariant MultiLayer Perceptron, which implies universality of symmetric MLPs and the FermiNet.

## 1 INTRODUCTION

Neural Networks (NN), or more precisely, Multi-Layer Perceptrons (MLP), are universal function approximators [Pin99] in the sense that every (say) continuous function can be approximated arbitrarily well by a sufficiently large NN. The true power of NN though stems from the fact that they apparently have a bias towards functions we care about and that they can be trained by local gradient-descent or variations thereof.

For many problems we have additional information about the function, e.g. symmetries under which the function of interest is invariant or covariant. Here we consider functions that are covariant[1] under permutations.[2] Of particular interest are functions that are invariant[3], equivariant[4], or anti-symmetric[5] under permutations.

**Definition 1 ((Anti)symmetric and equivariant functions)** *A function* $\phi : \mathcal{X}^n \to \mathbb{R}$ *in* $n \in \mathbb{N}$ *variables is called* symmetric *iff* $\phi(x_1, ..., x_n) = \phi(x_{\pi(1)}, ..., x_{\pi(n)})$ *for all* $x_1, ..., x_n \in \mathcal{X}$ *for all permutations* $\pi \in S_n$, *where* $S_n := \{\pi : \{1 : n\} \to \{1 : n\} \wedge \pi \text{ is bijection}\}$ *is called the symmetric group and* $\{1 : n\}$ *is short for* $\{1, ..., n\}$. *Similarly, a function* $\psi : \mathcal{X}^n \to \mathbb{R}$ *is called* anti-symmetric (AS) *iff* $\psi(x_1, ..., x_n) = \sigma(\pi)\psi(x_{\pi(1)}, ..., x_{\pi(n)})$, *where* $\sigma(\pi) = \pm 1$ *is the parity or sign of permutation* $\pi$. *A function* $\varphi : \mathcal{X}^n \to \mathcal{X}'^n$ *is called equivariant under permutations iff* $\varphi(S_\pi(\mathbf{x})) = S_\pi(\varphi(\mathbf{x}))$, *where* $\mathbf{x} \equiv (x_1, ..., x_n)$ *and* $S_\pi(x_1, ..., x_n) := (x_{\pi(1)}, ..., x_{\pi(n)})$.

Of course (anti)symmetric functions are also just functions, hence a NN of sufficient capacity can also represent (anti)symmetric functions, and if trained on an (anti)symmetric target could converge to an (anti)symmetric function. But NNs that can represent *only* (anti)symmetric functions are desirable for multiple reasons. Equivariant MLP (EMLP) are the basis for constructing symmetric functions by simply summing the output of the last layer, and for anti-symmetric (AS) functions by

---

[1] In full generality, a function $f : \mathcal{X} \to \mathcal{Y}$ is covariant under group operations $g \in G$, if $f(R_g^X(x)) = R_g^Y(f(x))$, where $R_g^X : \mathcal{X} \to \mathcal{X}$ and $R_g^Y : \mathcal{Y} \to \mathcal{Y}$ are representations of group (element) $g \in G$.

[2] The symmetric group $G = S_n$ is the group of all permutations=bijections $\pi : \{1, ..., n\} \to \{1, ..., n\}$.

[3] $R_g^Y$=Identity. Permutation-invariant functions are also called 'totally symmetric functions' or simply 'symmetric function'.

[4] General $\mathcal{Y}$ and $\mathcal{X}$, often $\mathcal{Y} = \mathcal{X}$ and $R_g^Y = R_g^X$, also called *covariant*.

[5] $R_g^Y = \pm 1$ for even/odd permutations.

multiplying with Vandermonde determinants or by computing their generalized Slater determinant (GSD) defined later.

The most prominent application is in quantum physics which represents systems of identical (fermions) bosons with (anti)symmetric wave functions [PSMF20]. Another application is classification of point clouds in computer vision, which should be invariant under permutation of points [ZKR$^+$18].

Even if a general NN can learn the (anti)symmetry, it will only do so approximately, but some applications require exact (anti)symmetry, for instance in quantum physics to guarantee upper bounds on the true ground state energy [PSMF20]. This has spawned interest in NNs that can represent *only* (anti)symmetric functions [ZKR$^+$18, HLL$^+$19]. A natural question is whether such NNs can represent *all* reasonable (anti)symmetric functions, which is the focus of this paper. We will answer this question for the (symmetric) EMLP [ZKR$^+$18] defined in Section 6 and for the (AS) FermiNet [PSMF20] defined in Sections 4&5&6.

Approximation architectures need to satisfy a number of criteria to be practically useful:

(a) they can approximate a large class of functions,
    e.g. all continuous (anti)symmetric functions,
(b) *only* the (anti)symmetric functions can be represented,
(c) a fast algorithm exists for computing the approximation,
(d) the representation itself is continuous or differentiable,
(e) the architecture is suitable for learning the function from data
    (which we don't discuss).

Section 2 reviews existing approximation results for (anti)symmetric functions. Section 3 discusses various "naive" representations (linear, sampling, sorting) and their (dis)advantages, before introducing the "standard" solution that satisfies (a)-(e) based on algebraic composition of basis functions, symmetric polynomials, and polarized bases. For simplicity the section considers only totally symmetric functions of their $n$ real-valued inputs (the $d = 1$ case), i.e. particles in one dimension. Section 4 proves the representation power of a single GSD for totally anti-symmetric (AS) functions (also $d = 1$). Technically we reduce the GSD to a Vandermonde determinant, and determine the loss of differentiability due to the Vandermonde determinant. From Sections 5 on we consider the general case of functions with $n \cdot d$ inputs that are (anti)symmetric when permuting their $n$ $d$-dimensional input vectors. The case $d = 3$ is particularly relevant for particles and point clouds in 3D space. The difficulties encountered for $d = 1$ transfer to $d > 1$, while the positive results don't, or only with considerable extra effort. The universality construction and proof for the EMLP is outlined in Section 6 with a proper treatment and all details in Sections 6-8 of the supplementary, which implies universality of symmetric MLPs and of the AS FermiNet. Section 7 concludes. We took great care to unify notation from different sources. The list of notation in the appendix should be helpful to disambiguate some similarly looking but different notation.

Our main novel contributions are establishing the universality of the anti-symmetric FermiNet with a single GSD (Theorems 3&5&7) for $d = 1$ and $d > 1$ (the results are non-trivial and unexpected), and the universality of (2-hidden-layer) symmetric MLPs (Theorem 6) with a complete and explicit and self-contained equivariant universality construction based on (smooth) polynomials. We took care to avoid relying on results with inherently asymptotic or tabulation or discontinuous character, to enable (in future work) good approximation rates for specific function classes, such as smooth functions or those with 'nice' Fourier transform [Bar93, Mak96], The supplementary material contains the extended version of this paper with (more) details, discussion, and proofs.

## 2 RELATED WORK

The study of universal approximation properties of NN has a long history, see e.g. [Pin99] for a pre-millennium survey, and e.g. [LSYZ20] for recent results and references. For (anti)symmetric NN such investigation has only recently begun [ZKR$^+$18, WFE$^+$19, HLL$^+$19, SI19].

Functions on sets are necessarily invariant under permutation, since the order of set elements is irrelevant. For countable domain, [ZKR$^+$18] derive a general representation based on encoding domain elements as bits into the binary expansion of real numbers. They conjecture that the construction

can be generalized to uncountable domains such as $\mathbb{R}^d$, but it would have to involve pathological everywhere discontinuous functions [WFE$^+$19]. Functions on sets of fixed size $n$ are equivalent to symmetric functions in $n$ variables. [ZKR$^+$18] prove a symmetric version of Kolmogorov-Arnold's superposition theorem [Kol57] (for $d = 1$) based on elementary symmetric polynomials und using Newton's identities, also known as Girard-Newton or Newton-Girard formulae, which we will generalize to $d > 1$. Another proof is provided based on homeomorphisms between vectors and ordered vectors, also with no obvious generalization to $d > 1$. They do not consider AS functions.

For symmetric functions and any $d \geq 1$, [HLL$^+$19] provide two proofs of the symmetric superposition theorem of [ZKR$^+$18]: Every symmetric function can be approximated by symmetric polynomials, symmetrized monomials can be represented as a permanents, and Ryser's formula brings the representation into the desired polarized superposition form. The down-side is that computing permanents is NP complete, and exponentially many symmetrized monomials are needed to approximate $f$. The second proof discretizes the input space into a $n \cdot d$-dimensional lattice and uses indicator functions for each grid cell. They then symmetrize the indicator functions, and approximate $f$ by these piecewise constant symmetric indicator functions instead of polynomials, also using Ryser formula for the final representation. Super-exponentially many indicator functions are needed, but explicit error bounds are provided. The construction is discontinuous but they remark on how to make it continuous. Approximating AS $f$ for $d \geq 1$ is based on a similar lattice construction, but by summing super-exponentially many Vandermonde determinants, leading to a similar bound. We show that a single Vandermonde/Slater determinant suffices but without bound. Additionally for $d = 1$ we determine the loss in smoothness this construction suffers from.

[SI19] prove tighter but still exponential bounds if $f$ is Lipschitz w.r.t. $\ell^\infty$ based on sorting which inevitably introduces irreparable discontinuities for $d > 1$.

The FermiNet [PSMF20] is also based on EMLPs [ZKR$^+$18] but anti-symmetrizes not with Vandermonde determinants but with GSDs. It has shown remarkable practical performance for modelling the ground state of a variety of atoms and small molecules. To achieve good performance, a linear combination of GSDs has been used. We show that in principle a single GSD suffices, a sort of generalized Hartree-Fock approximation. This is contrast to the increasing number of conventional Slater determinants required for increasing accuracy. Our result implies (with some caveats) that the improved practical performance of multiple GSDs is due to a limited (approximation and/or learning) capacity of the EMLP, rather than a fundamental limit of the GSD.

## 3 ONE-DIMENSIONAL SYMMETRY

This section reviews various approaches to representing symmetric functions, and is the broadest review we are aware of. To ease discussion and notation, we consider $d = 1$ in this section. Most considerations generalize easily to $d > 1$, some require significant effort, and others break. We discuss various "naive" representations (linear, sampling) and their (dis)advantages, before introducing the "standard" solution that can satisfy (a)-(e). All representations consist of a finite set of fixed (inner) basis functions, which are linearly, algebraically, functionally, or otherwise combined. We then introduce symmetric polynomials, which can be used to prove the "standard" representation theorem for $d = 1$.

The extended version contains a broader and deeper review of alternative representations, including composition by inversion, generally invariant linear bases, symmetric functions by sorting, and linear bases for symmetric polynomials. Indeed it is the broadest review we are aware of, and unified and summarized as far as possible in one big table. The extended review may also help to better grasp the concepts introduced in this section, since it is less dense and contains some illustrating examples.

**Motivation.** Consider $n \in \mathbb{N}$ one-dimensional particles with coordinates $x_i \in \mathbb{R}$ for particle $i = 1, ..., n$. In quantum mechanics the probability amplitude of the ground state can be described by a real-valued joint wave function $\chi(x_1, ..., x_n)$. Bosons $\phi$ have a totally symmetric wave function: $\phi(x_1, ..., x_n) = \phi(x_{\pi(1)}, ..., x_{\pi(n)})$ for all permutations $\pi \in S_n \subset \{1 : n\} \to \{1 : n\}$. Fermions $\psi$ have totally Anti-Symmetric (AS) wave functions: $\psi(x_1, ..., x_n) = \sigma(\pi)\psi(x_{\pi(1)}, ..., x_{\pi(n)})$, where $\sigma(\pi) = \pm 1$ is the parity or sign of permutation $\pi$. Wave functions are continuous and almost everywhere differentiable, and often posses higher derivatives or are even analytic. Nothing in this

work hinges on any special properties wave functions may possess or interpreting them as such, and the precise conditions required for our results to hold are stated in the theorems.

We are interested in representing or approximating all and only such (anti)symmetric functions by neural networks. Abbreviate $\mathbf{x} \equiv (x_1, ..., x_n)$ and let $S_\pi(\mathbf{x}) := (x_{\pi(1)}, ..., x_{\pi(n)})$ be the permuted coordinates. There is an easy way to (anti)symmetrize any function,

$$\phi(\mathbf{x}) \;=\; \frac{1}{n!} \sum_{\pi \in S_n} \chi(S_\pi(\mathbf{x})), \quad \psi(\mathbf{x}) \;=\; \frac{1}{n!} \sum_{\pi \in S_n} \sigma(\pi) \chi(S_\pi(\mathbf{x})) \tag{1}$$

and any (anti)symmetric function can be represented in this form (proof: use $\chi := \phi$ or $\chi := \psi$). If we train a NN $\chi : \mathbb{R}^n \to \mathbb{R}$ to approximate some function $f : \mathbb{R}^n \to \mathbb{R}$ to accuracy $\varepsilon > 0$, then $\phi$ ($\psi$) are (anti)symmetric approximations of $f$ to accuracy $\varepsilon > 0$ too, provided $f$ itself is (anti)symmetric. Instead of averaging, the minimum or maximum or median or many other compositions would also work, but the average has the advantage that smooth $\chi$ lead to smooth $\phi$ and $\psi$, and more general, preserves many desirable properties such as (Lipschitz/absolute/...) continuity, ($k$-times) differentiability, analyticity, etc. It possibly has all important desirable properties, but one:

**Time complexity, sampling, learning.** The problem with this approach is that it has $n!$ terms, and evaluating $\chi$ super-exponentially often is intractable even for moderate $n$, especially if $\chi$ is a NN. There can also be no clever trick to linearly (anti)symmetrize arbitrary functions fast, intuitively since the sum pools $n!$ independent regions of $\chi$. In the extended version we prove that computing $\phi$ and $\psi$ are indeed NP-hard. There we also show that approximating (1) by sampling permutations is unsuitable, especially for $\psi$ due to sign cancellations. Even if we could compute (1) fast, a NN would represent the function separately on all $n!$ regions, hence potentially requires $n!$ more training samples to learn from than an intrinsically (anti)symmetric NN. See the extended version for a more detailed discussion.

**Function composition and bases.** Before delving into proving universality of the EMLP and the FermiNet, it is instructive to first review the general concepts of function composition and basis functions, since a NN essentially is a composition of basis functions. We want to represent/decompose functions as $f(\mathbf{x}) = g(\boldsymbol{\beta}(\mathbf{x}))$. In this work we are interested in symmetric $\boldsymbol{\beta}$, where ultimately $\boldsymbol{\beta}$ will be represented by the first (couple of) layer(s) of an EMLP, and $g$ by the second (couple of) layer(s). Of particular interest is

$$\boldsymbol{\beta}(\mathbf{x}) = \sum_{i=1}^n \boldsymbol{\eta}(x_i) \tag{2}$$

for then $\boldsymbol{\beta}$ and hence $f$ are obviously symmetric (permutation invariant) in $\mathbf{x}$. Anti-symmetry is more difficult and will be dealt with later. Formally let $f \in \mathcal{F} \subseteq \mathbb{R}^n \to \mathbb{R}$ be a function (class) we wish to represent or approximate. Let $\beta_b : \mathbb{R}^n \to \mathbb{R}$ be basis functions for $b = 1, ..., m \in \mathbb{N} \cup \{\infty\}$, and $\boldsymbol{\beta} \equiv (\beta_1, ..., \beta_m) : \mathbb{R}^n \to \mathbb{R}^m$ be what we call basis vector (function), and $\eta_b : \mathbb{R} \to \mathbb{R}$ a basis template, sometimes called inner function [Act18] or polarized bass function. Let $g \in \mathcal{G} \subseteq \mathbb{R}^m \to \mathbb{R}$ be a composition function (class), sometimes called 'outer function' [Act18], which creates new functions from the basis functions. Let $\mathcal{G} \circ \boldsymbol{\beta} = \{g(\boldsymbol{\beta}(\cdot)) : g \in \mathcal{G}\}$ be the class of representable functions, and $\overline{\mathcal{G} \circ \boldsymbol{\beta}}$ its topological closure, i.e. the class of all approximable functions.[①] $\boldsymbol{\beta}$ is called a $\mathcal{G}$-basis for $\mathcal{F}$ if $\mathcal{F} = \mathcal{G} \circ \boldsymbol{\beta}$ or $\mathcal{F} = \overline{\mathcal{G} \circ \boldsymbol{\beta}}$, depending on context. Interesting classes of compositions are linear $\mathcal{G}_{lin} := \{g : g(\mathbf{x}) = a_0 + \sum_{i=1}^m x_i; \; a_0, a_i \in \mathbb{R}\}$, algebraic $\mathcal{G}_{alg} := \{\text{multivariate polynomials}\}$, functional $\mathcal{G}_{func} := \mathbb{R}^m \to \mathbb{R}$, and $\mathcal{C}^k$-functional $\mathcal{G}_{func}^k := \mathcal{C}^k$ for $k$-times continuously differentiable functions. The extended version illustrates on some simple examples how larger composition classes $\mathcal{G}$ allow (drastically) smaller bases ($m$) to represent the same functions $\mathcal{F}$.

**Algebraic basis for symmetric polynomials.** It is well-known that the elementary symmetric polynomials $e_b(\mathbf{x})$ generated by

$$\prod_{i=1}^n (1 + \lambda x_i) \;=: \; 1 + \lambda e_1(\mathbf{x}) + \lambda^2 e_2(\mathbf{x}) + ... + \lambda^n e_n(\mathbf{x}) \tag{3}$$

---

[①] Functions may be defined on sub-spaces of $\mathbb{R}^k$, function composition may not exists, and convergence can be w.r.t. different topologies. We will ignore these technicalities unless important for our results, but the reader may assume compact-open topology, which induces uniform convergence on compacta.

are an algebraic basis of all symmetric polynomials. Explicit expressions are $e_1(\mathbf{x}) = \sum_i x_i$, and $e_2(\mathbf{x}) = \sum_{i<j} x_i x_j$, ..., and $e_n(\mathbf{x}) = x_1 ... x_n$, and in general $e_b(\mathbf{x}) = \sum_{i_1 < ... < i_b} x_{i_1} ... x_{i_b}$. For given $\mathbf{x}$, the polynomial in $\lambda$ on the l.h.s. of (3) can be expanded to the r.h.s. in quadratic time or by FFT even in time $O(n \log n)$, so the $\mathbf{e}(\mathbf{x})$ can be computed in time $O(n \log n)$, but is not of the desired form (2). Luckily Newton already solved this problem for us. Newton's identities express the elementary symmetric polynomials $e_1(\mathbf{x}), ..., e_n(\mathbf{x})$ as polynomials in $p_b(\mathbf{x}) := \sum_{i=1}^{n} x_i^b$, $b = 1, ..., n$, hence also $\boldsymbol{\beta}(\mathbf{x}) := (p_1(\mathbf{x}), ..., p_n(\mathbf{x}))$ is an algebraic basis for all symmetric polynomials, hence by closure for all continuous symmetric functions, and is of desired form (2):

**Theorem 2 (Symmetric polarized superposition [ZKR+18, WFE+19, Thm.7])** *Every continuous symmetric function* $\phi : \mathbb{R}^n \to \mathbb{R}$ *can be represented as* $\phi(\mathbf{x}) = g(\sum_i \boldsymbol{\eta}(x_i))$ *with* $\boldsymbol{\eta}(x) = (x, x^2, ..., x^n)$ *and continuous* $g : \mathbb{R}^n \to \mathbb{R}$.

[ZKR+18] provide two proofs, one based on 'composition by inversion', the other using symmetric polynomials and Newton's identities. The non-trivial generalization to $d > 1$ is provided in Section 5.

Theorem 2 is a symmetric version of the infamous Kolmogorov-Arnold superposition theorem [Kol57], which solved Hilbert's 13th problem. Its deep and obscure[7] constructions continue to fill whole PhD theses [Liu15, Act18]. It is quite remarkable that the symmetric version above is very natural and comparably easy to prove.

For given $\mathbf{x}$, the basis $\boldsymbol{\beta}(\mathbf{x})$ can be computed in time $O(n^2)$, so is actually slower to compute than $\mathbf{e}(\mathbf{x})$. The elementary symmetric polynomials also have other advantages (integral coefficients for integral polynomials, works for fields other than $\mathbb{R}$, is numerically more stable, mimics 1,2,3,... particle interactions), so symmetric NN based on $e_b$ rather than $p_b$ may be worth pursuing. Note that we need at least $m \geq n$ functional bases for a *continuous* representation, so Theorem 2 is optimal in this sense [WFE+19]. The extended version contains a discussion and a table with bases and properties.

## 4 ONE-DIMENSIONAL ANTISYMMETRY

We now consider the anti-symmetric (AS) case for $d = 1$. We provide representations of AS functions in terms of generalized Slater determinants (GSD) of *partially* symmetric functions. In later sections we will discuss how these partially symmetric functions arise from equivariant functions and how to represent equivariant functions by EMLP. The reason for deferral is that EMLP are inherently tied to $d > 1$. Technically we show that the GSD can be reduced to a Vandermonde determinant, and exhibit a potential loss of differentiability due to the Vandermonde determinant.

Let $\varphi_i : \mathbb{R} \to \mathbb{R}$ be single-particle wave functions. Consider the matrix

$$\Phi(\mathbf{x}) = \begin{pmatrix} \varphi_1(x_1) & \cdots & \varphi_n(x_1) \\ \vdots & \ddots & \vdots \\ \varphi_1(x_n) & \cdots & \varphi_n(x_n) \end{pmatrix}$$

where $\mathbf{x} \equiv (x_1, ..., x_n)$. The (Slater) determinant $\det \Phi(\mathbf{x})$ is anti-symmetric, but can represent only a small class of AS functions, essentially the AS analogue of product (wave) functions (pure states, Hartree-Fock approximation). Every continuous AS function can be approximated/represented by a finite/infinite linear combination of such determinants:

$$\psi(x_1, ..., x_n) = \sum_{k=1}^{\infty} \det \Phi^{(k)}(\mathbf{x}), \quad \text{where} \quad \Phi_{ij}^{(k)}(\mathbf{x}) := \varphi_i^{(k)}(x_j)$$

An alternative is to generalize the Slater determinant itself [PSMF20] by allowing the functions $\varphi_i(x_j)$ to depend on all variables

$$\Phi(\mathbf{x}) = \begin{pmatrix} \varphi_1(x_1|x_{\neq 1}) & \cdots & \varphi_n(x_1|x_{\neq 1}) \\ \vdots & \ddots & \vdots \\ \varphi_1(x_n|x_{\neq n}) & \cdots & \varphi_n(x_n|x_{\neq n}) \end{pmatrix}$$

---

[7]involving continuous $\eta$ with derivative 0 almost everywhere, and not differentiable on a dense set of points.

where $x_{\neq i} \equiv (x_1, ..., x_{i-1}, x_{i+1}, ..., x_n)$. If $\varphi_i(x_j|x_{\neq j})$ is symmetric in $x_{\neq j}$, which we henceforth assume[8], then exchanging $x_i \leftrightarrow x_j$ is (still) equivalent to exchanging rows $i$ and $j$ in $\Phi(\mathbf{x})$, hence $\det \Phi$ is still AS. The question arises how many GSD are needed to be able to represent *every* AS function $\psi$. The answer turns out to be 'just one', but with non-obvious smoothness relations: Any AS $\psi$ can be represented by some $\Phi$, any analytic $\psi$ can be represented by an analytic $\Phi$, for $k$-times continuously differentiable $\psi \in \mathcal{C}^k$, a large number of derivatives are *potentially* lost:

**Theorem 3 (Representation of all (analytic) AS $\psi$)** *For every (general/analytic/$\mathcal{C}^{k+n(n+1)/2}$) AS function $\psi(\mathbf{x})$ there exist (general/analytic/$\mathcal{C}^k$) $\varphi_i(x_j|x_{\neq j})$ symmetric in $x_{\neq j}$ such that $\psi(\mathbf{x}) = \det \Phi(\mathbf{x})$.*

**Proof. sketch.** The proof uses $\varphi_1(x_j|x_{\neq j}) := \chi(x_{1:n})$ with $\chi(\mathbf{x}) := \psi(\mathbf{x})/\Delta(\mathbf{x})$ and $\Delta(\mathbf{x}) := \prod_{j<i}(x_i - x_j)$ and $\varphi_i(x_j|x_{\neq j}) := x_j^{i-1}$ for $1 < i \leq d$. For this choice, $\det \Phi$ reduces to $\chi$ times the Vandermonde determinant $\Delta$. $\chi$ is obviously symmetric, but properly extending its definition to $\mathbf{x}$ for which $\Delta(\mathbf{x}) = 0$ is subtle. For general $\psi$, any continuation will do. For polynomial $\psi$, representation $\psi = \chi \cdot \Delta$ in terms of a symmetric polynomial $\chi$ and AS polynomial $\Delta$ is well-known. This can be extended to analytic $\psi$ via Taylor series expansion. For $k'$-times differentiable $\psi$, the construction is more difficult and the reduction in differentiability unfortunate and perhaps surprising. Full proofs can be found in the extended version. ∎

We do not know whether the loss of $\frac{1}{2}n(n+1)$ derivatives is an artefact of the proof or 'real'. For instance, we have seen that linear anti-symmetrization preserves differentiability. In Section 5 we show that continuity is preserved (only for $d = 1$).

## 5 $d$-DIMENSIONAL (ANTI)SYMMETRY

This section generalizes the theorems from Sections 3 and 4 to $d > 1$: the symmetric polynomial algebraic basis and the generalized Slater determinant representation.

**Motivation.** We now consider $n \in \mathbb{N}$, $d$-dimensional particles with coordinates $\boldsymbol{x}_i \in \mathbb{R}^d$ for particles $i = 1, ..., n$. For $d = 3$ we write $\boldsymbol{x}_i = (x_i, y_i, z_i)^\top \in \mathbb{R}^3$. As before, Bosons/Fermions have symmetric/AS wave functions $\chi(\boldsymbol{x}_1, ..., \boldsymbol{x}_n)$. That is, $\chi$ does not change/changes sign under the exchange of two vectors $\boldsymbol{x}_i$ and $\boldsymbol{x}_j$. It is *not* symmetric/AS under the exchange of individual coordinates e.g. $y_i \leftrightarrow y_j$. $\mathbf{X} \equiv (\boldsymbol{x}_1, ..., \boldsymbol{x}_n)$ is a matrix with $n$ columns and $d$ rows. The (representation of the) symmetry group is $\mathcal{S}_n^d := \{S_\pi^d : \pi \in S_n\}$ with $S_\pi^d(\boldsymbol{x}_1, ..., \boldsymbol{x}_n) := (\boldsymbol{x}_{\pi(1)}, ..., \boldsymbol{x}_{\pi(n)})$, rather than $\mathcal{S}_{n \cdot d}$. Functions $f : \mathbb{R}^{d \cdot n} \to \mathbb{R}$ invariant under $\mathcal{S}_n^d$ are sometimes called multisymmetric or block-symmetric, if calling them symmetric could cause confusion.

**Algebraic basis for multisymmetric polynomials.** The elementary symmetric polynomials (3) have a generalization to $d > 1$ [Wey46]. We only present them for $d = 3$. The general case is obvious from them. They can be generated from

$$\prod_{i=1}^n (1 + \lambda x_i + \mu y_i + \nu z_i) =: \sum_{0 \leq p+q+r \leq n} \lambda^p \mu^q \nu^r e_{pqr}(\mathbf{X}) \tag{4}$$

Even for $d = 3$ the explicit expressions for $e_{pqr}$ are rather cumbersome, but straightforward to obtain. One can show that $\{e_{pqr} : p+q+r \leq n\}$ is an algebraic basis of size $m = \binom{n+3}{3} - 1$ for all multisymmetric polynomials [Wey46]. Note that constant $e_{000}$ is not included/needed. For a given $\mathbf{X}$, their values can be computed in time $O(mn)$ by expanding (4) or in time $O(m \log n)$ by FFT, where $m = O(n^d)$. Newton's identities also generalize: $e_{pqr}(\mathbf{X})$ are polynomials in the polarized sums $p_{pqr}(\mathbf{X}) := \sum_{i=1}^n \eta_{pqr}(\boldsymbol{x}_i)$ with $\eta_{pqr}(\boldsymbol{x}) := x^p y^q z^r$. The proofs are much more involved than for $d = 1$. For the general $d$-case we have:

**Theorem 4 (Multisymmetric polynomial algebraic basis [Wey46])** *Every continuous (block-=multi)symmetric function $\phi : \mathbb{R}^{n \cdot d} \to \mathbb{R}$ can be represented as $\phi(\mathbf{X}) = g(\sum_{i=1}^n \boldsymbol{\eta}(\boldsymbol{x}_i))$ with continuous $g : \mathbb{R}^m \to \mathbb{R}$ and $\boldsymbol{\eta} : \mathbb{R}^d \to \mathbb{R}^m$ defined as $\eta_{p_1...p_d}(\boldsymbol{x}) = x^{p_1} y^{p_2} ... z^{p_d}$ for $1 \leq p_1 + ... + p_d \leq n$ ($p_i \in \{0, ..., n\}$), hence $m = \binom{n+d}{d} - 1$.*

---

[8] The bar | is used to visually indicate this symmetry, otherwise there is no difference to using a comma.

The basis can be computed in time $O(m \cdot d) \subseteq O(d \cdot (n+1)^d)$. Note that there could be much smaller functional bases of size $m = dn$ as per "our" composition-by-inversion argument for continuous representations in Section 3 of the extended version, which readily generalizes to $d > 1$, while the above minimal algebraic basis has larger size $m = O(n^d)$ for $n \gg d > 1$. It is an open question whether a continuous functional basis of size $O(dn)$ exists, whether in polarized form (2) or not.

**Anti-Symmetry.** For $d = 1$, *all* AS functions $\psi$ have the *same* (core) zeros (called Fermion nodes), namely when $x_i = x_j$ for some $i \neq j$, which form a union of linear spaces dividing $\mathbb{R}^n$ into $n!$ isomorphic partitions on which $\psi$ are identical apart from sign $\pm 1$. This fact allowed representing every $\psi$ as a product of a symmetric function $\phi$ and the universal anti-symmetric polynomial $\Delta$, leading to representation Theorem 3. For $d > 1$, the Fermion nodes $\{\mathbf{X} : \psi(\mathbf{X}) = 0\}$ form essentially arbitrary $\psi$-dependent unions of (non-linear) manifolds partitioning $\mathbb{R}^{dn}$ into an arbitrary even number of cells of essentially arbitrary topology [Mit07]. This fact prevents a similar simple factoring ($\psi = \phi \cdot \Delta$) and Vandermonde-like reduction in the proof of Theorem 3, and indeed prevents finite *algebraic* bases for AS polynomials. We can still show a similar representation result, albeit weaker and via a different construction:

As in Section 4, consider $\Phi_{ij}(\mathbf{X}) := \varphi_i(\boldsymbol{x}_j | \boldsymbol{x}_{\neq j})$, i.e.

$$\Phi(\mathbf{X}) = \begin{pmatrix} \varphi_1(\boldsymbol{x}_1 | \boldsymbol{x}_{\neq 1}) & \cdots & \varphi_n(\boldsymbol{x}_1 | \boldsymbol{x}_{\neq 1}) \\ \vdots & \ddots & \vdots \\ \varphi_1(\boldsymbol{x}_n | \boldsymbol{x}_{\neq n}) & \cdots & \varphi_n(\boldsymbol{x}_n | \boldsymbol{x}_{\neq n}) \end{pmatrix}$$

where $\varphi_i(\boldsymbol{x}_j | \boldsymbol{x}_{\neq j})$ is symmetric in $\boldsymbol{x}_{\neq j}$.

**Theorem 5 (Representation of all AS $\psi$)** *For every AS function $\psi(\mathbf{X})$ there exist $\varphi_i(\boldsymbol{x}_j | \boldsymbol{x}_{\neq j})$ symmetric in $\boldsymbol{x}_{\neq j}$ such that $\psi(\mathbf{X}) = \det \Phi(\mathbf{X})$.*

The proof in the extended version is based on sorting $\boldsymbol{x}_{\bar{\pi}(1)} \leq \boldsymbol{x}_{\bar{\pi}(2)} \leq ... \leq \boldsymbol{x}_{\bar{\pi}(n)}$ with suitable $\bar{\pi}$, which is a discontinuous operation in $\mathbf{X}$ for $d > 1$. It is continuous though not differentiable for $d = 1$, hence $\varphi_i$ can be chosen continuous for continuous $\psi$ in $d = 1$. For $n = 2$ and any $d$, any AS continuous/smooth/analytic/$\mathcal{C}^k$ function $\psi(\boldsymbol{x}_1, \boldsymbol{x}_2)$ has an easy continuous/smooth/analytic/$\mathcal{C}^k$ representation as a GSD. Choose $\varphi_1(\boldsymbol{x}_1 | \boldsymbol{x}_2) := \frac{1}{2}$ and $\varphi_2(\boldsymbol{x}_1 | \boldsymbol{x}_2) := \psi(\boldsymbol{x}_1, \boldsymbol{x}_2)$. Whether this generalizes to $n > 2$ and $d > 1$ is an open problem.

# 6 NEURAL NETWORKS

In this section we will restrict the representation power of classical MLPs to equivariant functions, which are then used to define universal (anti)symmetric NN.

**Equivariance and all-but-one symmetry.** We are mostly interested in (anti)symmetric functions, but for (de)composition we need equivariant functions, and directly need to consider the $d$-dimensional case. A function $\boldsymbol{\varphi} : (\mathbb{R}^d)^n \to (\mathbb{R}^{d'})^n$ is called equivariant under permutations if $\boldsymbol{\varphi}(S_\pi^d(\mathbf{X})) = S_\pi^d(\boldsymbol{\varphi}(\mathbf{X}))$ for all permutations $\pi \in S_n$. With slight abuse of notation we identify $(\boldsymbol{\varphi}(\mathbf{X}))_1 \equiv \varphi_1(\mathbf{X}) \equiv \varphi_1(\boldsymbol{x}_1, \boldsymbol{x}_2, ..., \boldsymbol{x}_n) \equiv \varphi_1(\boldsymbol{x}_1, \boldsymbol{x}_{\neq 1})$ with $\varphi_1(\boldsymbol{x}_1 | \boldsymbol{x}_{\neq 1})$. It is easy to see (see extended version) that a function $\boldsymbol{\varphi}$ is equivariant (under permutations) if and only if $\varphi_i(\mathbf{X}) = \varphi_1(\boldsymbol{x}_i | \boldsymbol{x}_{\neq i})$ $\forall i$ and $\varphi_1(\boldsymbol{x}_i | \boldsymbol{x}_{\neq i})$ is symmetric in $\boldsymbol{x}_{\neq i}$. Hence $\boldsymbol{\varphi}_1$ suffices to describe all of $\boldsymbol{\varphi}$.

**Equivariant Neural Network.** We aim at approximating equivariant $\boldsymbol{\varphi}$ by an Equivariant MLP (EMLP) defined as follows: The output $\mathbf{X}' \in \mathbb{R}^{d' \times n}$ of an EMLP layer is computed from its input $\mathbf{X} \in \mathbb{R}^{d \times n}$ by

$$\boldsymbol{x}_i' := \tau_i(\mathbf{X}) := \tau_1(\boldsymbol{x}_i | \boldsymbol{x}_{\neq i}) \equiv \tau_{1, \mathbf{W}, \mathbf{V}, \boldsymbol{u}}(\boldsymbol{x}_i | \boldsymbol{x}_{\neq i}) := \boldsymbol{\sigma}(\mathbf{W}\boldsymbol{x}_i + \mathbf{V}\textstyle\sum_{j \neq i} \boldsymbol{x}_j + \boldsymbol{u}) \quad (5)$$

where $\boldsymbol{\tau} : \mathbb{R}^{d \times n} \to \mathbb{R}^{d' \times n}$ can be shown to be an equivariant "transfer" function with weight matrices $\mathbf{W}, \mathbf{V} \in \mathbb{R}^{d' \times d}$, and $\boldsymbol{u} \in \mathbb{R}^{d'}$ the biases. Using $\sum_j$ as in [ZKR+18] instead of $\sum_{j \neq i}$ as in [PSMF20] works as well. A $L$-(hidden)-layer EMLP concatenates $L$ such layers (with non-linearity removed from the last layer). It is easy to see that EMLP is indeed equivariant and that the argument of $\boldsymbol{\sigma}()$ is the only linear function in $\mathbf{X}$ with such invariance [ZKR+18, Lem.3].

In previous work it may have been tacitly been assumed that the universality of the polarized representation in Theorem 2 implies that EMLP can approximate *all* equivariant functions. In Sections 7&8 of the extended version we provide an explicit construction and proof based on Theorem 4. The explicit steps in the proof can be retraced if one wishes to derive error bounds.

**Theorem 6 (Universality of (two-hidden-layer) EMLP)** *For any continuous non-linear activation function, EMLPs can approximate (uniformly on compacta) all and only the equivariant continuous functions. If $\sigma$ is non-polynomial, a two-hidden-layer EMLP suffices.*

**Proof. idea.** The construction follows 4 steps: (1) representation of polynomials in a single vector $\boldsymbol{x}_i$, (2) multisymmetric polynomials in all-but-one vector crucially exploiting Theorem 4, (3) equivariant polynomials, (4) equivariant continuous functions. Each step (1-3) constructs an EMLP. One hidden-layer EMLP suffice by classical results for MLPs [Pin99]. Two layers can be merged, leading to a 2-hidden-layer EMLP. Step (4) uses the (Stone-)Weierstrass theorem, for which also quantitative versions exist with error bounds, e.g. based on Chebyshev polynomials. ∎

**Universal Symmmetric Network.** We can approximate all symmetric continuous functions by applying any symmetric continuous function $\varsigma : \mathbb{R}^{d' \times n} \to \mathbb{R}^{d'}$ with the property $\varsigma(\boldsymbol{y}, ..., \boldsymbol{y}) = \boldsymbol{y}$ to the output of an EMLP, e.g. $\varsigma(\mathbf{Y}) = \frac{1}{n}\sum_{i=1}^{n} \boldsymbol{y}_i$ or $\varsigma(\mathbf{Y}) = \max\{y_1, ..., y_n\}$ if $d' = 1$.

**Universal AntiSymmmetric Network.** By Theorem 5 we know that every AS function $\psi$ can be represented as a GSD of $n$ functions symmetric in all-but-one-variable. The proof of Theorem 5 requires only a single symmetric function, but using an EMLP with $n$ equivariant function can potentially preserve smoothness as discussed in the extended paper.

Let us define a (toy) FermiNet as computing a single GSD from the output of a universal EMLP. The real FermiNet developed in [PSMF20] contains a number of extra features, which improves practical performance, theoretically most notably particle pair representations. Since it is a superset of our toy definition, the following theorem also applies to the full FermiNet. We arrived at the following result (under the same conditions as in Theorem 6):

**Theorem 7 (Universality of the FermiNet)** *A FermiNet with a single GSD can approximate any continuous anti-symmetric function.*

For $d = 1$, the approximation is again uniform on compacta. For $d > 1$, the proof of Theorem 5 involves discontinuous $\varphi_i(\boldsymbol{x}_j|\boldsymbol{x}_{\neq j})$. Any discontinuous function can be approximated by continuous functions, not in $\infty$-norm but only weaker $p$-norm for $1 \leq p < \infty$. This implies the theorem also holds for $d > 1$ in $L^p$ norm. Whether a stronger $L^{\infty}$ result holds is an important open problem, important because approximating continuous functions by discontinuous components can cause all kinds of problems.

# 7 Discussion

We reviewed a variety of representations for (anti)symmetric functions $(\psi)\phi : (\mathbb{R}^d)^n \to \mathbb{R}$. The most direct and natural way is as a sum over $n!$ permutations of some other function $\chi$. If $\chi \in \mathcal{C}^k$ then also $(\psi)\phi \in \mathcal{C}^k$. Unfortunately this takes exponential time, or at least is NP hard, and other direct approaches such as sampling or sorting have their own problems. The most promising approach is using Equivariant MLPs, for which we provided a constructive and complete universality proof, combined with a trivial symmetrization and a non-trivial anti-symmetrization using a large number Slater determinants. We investigated to which extent a *single generalized* Slater determinant introduced in [PSMF20], which can be computed in time $O(n^3)$, can represent all AS $\psi$. We have shown that for $d = 1$, all AS $\psi \in \mathcal{C}^{k+n(n-1)/2}$ can be represented as $\det \Phi$ with $\Phi \in \mathcal{C}^k$. Whether $\Phi \in \mathcal{C}^k$ suffices to represent all $\psi \in \mathcal{C}^k$ is unknown for $k > 0$. For $k = 0$ it suffices. For $d > 1$ and $n > 2$, we were only able to show that AS $\psi$ have representations using discontinuous $\Phi$.

Important problems regarding smoothness of the representation are open in the AS case. Whether continuous $\Phi$ can represent all continuous $\psi$ is unknown for $d > 1$, and similar for differentiability and for other properties. Indeed, whether *any* computationally efficient continuous representation of all and only AS $\psi$ is possible is unknown.

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
