# OpenReview forum: "On Representing (Anti)Symmetric Functions"
_ICLR.cc/2021/Conference — Reject_

### Official Review · AnonReviewer1 · 2020-10-26
**Interesting topic but missed significant previous works**

**Rating:** 4
**Confidence:** 4

**Review:**

Post discussion:
I read the author's response and other reviews. I will stick to my rating and encourage the author to resubmit a revised version focusing on the antisymmetric case.





Summary:

The paper studies the approximation power and shows universality results for two recent neural network models that represent symmetric and antisymmetric functions. The main contributions of the paper are claimed to be (1) Universality of Fermi-Net (antisymmetric model) with a single Generalized Slater determinant (2) Universality of symmetric MLPs. In both cases, the authors emphasize the fact that the theorems deal with (i) vector inputs rather than scalar inputs, and that (ii) the approximation results are based on smooth polynomials rather than discontinuous functions as done in previous works.

While the problem the paper targets is interesting and important, the paper missed several important previous works and does not do a good job explaining their novelty with respect to the previous work that was cited. See below.

Contribution 1: I am not an expert on antisymmetric function approximation and didn’t know Ferminet, but contribution 1 seems novel to me, although the authors should make a better job explaining the difference between their results and the original FermiNet paper. It is not entirely clear what was done before and what is new

Contribution 2 and points (i),(ii) were discussed before in several papers. First “Provably Powerful Graph Networks” (NeurIPS 2019) used the power sum multi symmetric polynomials for representing set functions in the same way they are used in this paper.  Second, “On universal equivariant set networks” (ICLR 2020) proves a universal approximation theorem for equivariant set functions based on these polynomials. Given these two works, I am not sure the current paper has any additional contribution.

Strong points:

Understanding the approximation power of invariant/equivariant neural networks is an important goal.
The results on antisymmetric functions are nice

Weak points:

The work is not properly positioned with respect to previous work and pretty much misses all the work done on the approximation of invariant functions since DeepSets. Except for the discussion above, the following should be cited/discussed:

-- “Janossy Pooling: Learning Deep Permutation-Invariant Functions for Variable-Size Inputs” ICLR 2019, which discusses permutation sampling strategies.

-- “Universal approximations of invariant maps by neural networks” 2018 that discusses symmetrization and approximation of symmetric functions (and function invariant to many other compact groups).

-- “PointNet: Deep Learning on Point Sets for 3D Classification and Segmentation” CVPR 2017, which also show universal approximation for symmetric functions.

-- “On the Universality of Invariant Networks”, "Universal Equivariant Multilayer Perceptrons" ICML 2019/2020 might also be relevant.

Recommendation:

The paper studies an important problem but missed significant previous work. I believe that the results on antisymmetric function approximation are novel and suggest rewriting the paper focused on these results, with a clear discussion on the contribution with respect to previous works. It might also be good to spend more time on Ferminet while doing so since this model is less known to the machine learning community. In its current form, The paper is not ready for publication.

Minor comments:

The authors added a long version (22 pages) as an appendix. Not sure if this is OK.

---

> ### Author Response · Authors · 2020-11-20
> **Difference to FermiNet Paper and Missing References**
>
> Thank you for your review. The numbering below refers to the review’s numbering:
>
> C1) [PSMF20] introduces the FermiNet and experimentally evaluates it. They observe that increasing the number of GSDs increases empirical performance. There is no theoretical analysis in this paper. A natural and important question arises whether fewer GSDs even if compensated by larger NNs perform worse due to an inherent minimal approximation error, so a large number of GSDs is unavoidable, or for other reasons, e.g. failure of the learning algorithm, which if overcome could reduce the number of expensive GSD evaluations. The theoretically clean version of this question is whether a FermiNet with a bounded number of GSDs (or even just 1 GSD) is universal, or whether increasing accuracy requires increasing the number of GSDs. Since the latter is true for (a) classical Slater determinant representations and (b) experimentally for GSDs, the discovery that 1 GSD suffices in theory was unexpected (at least for d>1).
>
> C2) Thank you very much for the provided references. They are all sufficiently relevant to include in the background section, but the one true embarrassing omission is “On universal equivariant set networks” (ICLR 2020). This (great) paper indeed proves universality of EMLPs with essentially the same approach, though my polynomial construction is a little simpler. It is a bit surprising that it didn’t come up during my literature search. Given this prior work, the EMLP part of my submission becomes more-or-less redundant, leaving the anti-symmetric part (and the IMO nice survey of approaches in the supplementary), which, I admit, in itself doesn’t seem to warrant acceptance at ICLR unless someone could solve the loss-of-differentiability problem. The only consolation is that the statement by some of the other reviewers that (the proof of) Theorem 6 is trivial / straightforward / not worth publishing is at odds with the fact that the above work, which essentially is “just” this proof, has been accepted at ICLR.

---

### Official Review · AnonReviewer2 · 2020-10-27
**review for antisymmetric**

**Rating:** 4
**Confidence:** 4

**Review:**

In this paper the authors study the representability of symmetric or antisymmetric functions using neural networks. In particular, we say f: R^n -> R is symmetric/ antisymmetric if f(x)= f(pi(x)) for all pi in S_n or is of the form f(x)= sign(pi) *f(pi(x)) where sign(pi) is the sign of the permutation. Such functions with such symmetries are ubiquitous in quantum physics since if one looks at the wave function of a  system of identical bosons, then they are symmetric and if you look at fermions they are antisymmetric. In this paper they authors try to understand if there exists succinct architectures that can learn/represent functions with such a symmetry property.  In this direction the author makes a few observations For every "nice" antisymmetric function f, there exists a symmetric function Phi and a matrix associated to Phi such that determinant of Phi can be used to represent f.  Then the authors show that the standard Ferminet (the first demonstration of deep learning) can be used to represent determinants of slater determinant and hence an arbitrary continous antisymmetric function.

Overall, this paper is interesting to read but I wouldn't recommend it for acceptance for the following reasons:

1) There are many points which seem like "overselling". The author claims that a polynomial f: R^n -> R has n! many terms and that is too many to approximate and so on. Sure this is true, but the symmetries make a huge difference here. Additionally, there is a rich theory of approximation theory saying we can approximate symmetric functions by low-degree polynomials, so why not just allow the NN to compute this low-degree polynomial which in turn approximates f?

2) It seems to me that this paper is more of a review, than something novel. Even the author at many places claims that this paper is a thorough review, so its not clear if it clears the bar for ICLR.

3) The main results of this paper are straightforward in some sense. In fact the author doesn't even talk of the main results for the first 5 pages and the main results are simply swept under the rag, and it's not clear to me what is the novelty of the results in that case. The quantum motivation also seems very weak, i prefer a much more solid motivation for why such functions are useful (given the author mentions practicality).

Given these reservations I wouldn't accept it for recommendation.

---

> ### Author Response · Authors · 2020-11-20
> **Computation, Polynomial Approximation, Review, Straightforward**
>
> Thank you for your review. The numbering below refers to the review’s numbering
>
> 1) I think there is a misunderstanding here. If I use a standard MLP (or any architecture not respecting permutation symmetry), the only known way to convert this to a guaranteed and exactly (anti)symmetric function in a differentiable and universal way is by summing _all_ n! permutations as per Eq.(1). You write, “Sure this is true, but the symmetries make a huge difference here.” Which symmetries are you referring to? If χ is a general MLP with randomly initialized weights it contains no symmetries.  You may have in mind initializing or setting the weights to somehow make the MLP symmetric. Well, this is precisely the approach taken by [ZKR+18] and pursued here and proven to be universal in Theorem 6.
> Regarding low-degree polynomials: Of course you can approximate (anti)symmetric functions by general polynomials, but these approximations will in general not be _exactly_ (anti)symmetric, and exact (anti)symmetry is crucial in quantum physics applications. If we are content with approximate anti-symmetry, we could indeed just use a general MLPs, which we already know to be universal. But this may require an exponentially larger training size, as compared to using MLPs (EMLP/FermiNet) that are intrinsically (anti)symmetric. All this is explained in Section 3 of the paper.
>
> 2) The “broadest review” was intended to refer only to the extended version, but I realize that I failed to edit this out for the submission (which is not a review). Virtually everything introduced or discussed in the first 5 pages is either to motivate why simpler approaches fail -or- are required background or notation to formulate and understand the theorems and proofs, and rather compactly so. The extended version indeed contains a broad review, since apparently even the “overlong” background discussion in the main paper fails to confer why we need this complicated approach via Slater determinants, equivariant MLPs, and symmetric polynomials.
>
> 3) If Theorem 6 is straightforward, I am curious to hear whether universality holds for a 1-hidden layer EMLP, and if not, why 2 layers are needed while 1-hidden-layer MLPs are universal. If theorems Theorem 3 and 5 are straightforward, I am curious to hear whether the loss of differentiability in Theorem 3 is an artifact of the proof, or an intrinsic problem of the GSD, or, why the answers to these questions are not obvious. In hindsight and with sufficient knowledge, most proofs are “straightforward in some sense”. The main results are the stated theorems and discussed before and after the theorem and are not swept under the rug. The immense importance and practical success of the FermiNet is mentioned in Section 2 and demonstrated in [PSMF20]. I presume the reviewer would like me to add a paragraph about the importance of quantum physics simulations in general and [PSMF20] in particular.

---

### Official Review · AnonReviewer4 · 2020-10-28
**A good contribution to the understanding of representing symmetric and anti-symmetric functions**

**Rating:** 6
**Confidence:** 3

**Review:**

Update:
I have read the authors' responses and other comments.  I still think that the theoretical results on anti-symmetric functions of this submission are novel, which is, however, not well-delivered. A lot of space is wasted to discuss symmetric functions, for which the contributions of this paper are not clear.  I suggest substantially rewriting this paper by only focusing on the anti-symmetric functions.

===============================================================
Summary:

This paper studies the representation of symmetric and anti-symmetric functions as well as the parameterization with neural networks.  This issue is very important for applying deep learning to solve problems from computational quantum physics as well as point clouds.

Pros:

This paper is well-organized, in particular the connection to the previous work.  The representations of symmetric and anti-symmetric functions are very important but less-studied problems. As I understand it, this paper makes some interesting contributions to this topic.  Theorem 3 & 5 are very especially interesting, which basically shows that one Slater determinant is enough to approximate any anti-symmetric functions.

Cons:

While the theorems in this paper look pretty, it is not clear if they have computational advantages over the other ansatzes.  After all, no approximation rates and numerical results are provided. For example, computing Vandermonde determinants is much efficient than computing Slater determinants.  But as suggested in this paper, the approximation with Slater determinants is more efficient than the one with Vandermonde determinants. Combined them together, it is not clear which one will be more practical. Could you comment more on this point?


Other comments:
The authors do not put the proofs in the appendix.  Instead, they provide an extended version of the paper in the supplemental material which includes the proofs.  This makes the reading of the proof more challenging since readers have to dig out the proofs from the long paper. So basically, there are two versions of this paper.

---

> ### Author Response · Authors · 2020-11-20
> **Computational Advantages over other Ansatzes**
>
> Thank you for your review.
> Regarding your question about computational advantages: I am not sure which “other ansatzes” you are referring to. For d>1, and n>10 (say), the Slater determinant, which can be computed in time O(n^3), is the _only_ known computationally feasible approach for representing AS functions.
> For d=1 (only), AS functions can also be represented using the Vandermonde determinant, which can be computed in time O(n^2). Both are much faster than naive (anti)symmetrization, which takes time O(n!). Maybe you are referring to a sorting representation, which can be computed in time O(n), but this is not differentiable, or sampling which is not exact, both are problematic in quantum physics applications, so has any other approach that has been suggested so far. Sec.3 and Table 1 of the extended version reviews the difficulties alternative representation approaches face.

---

### Official Review · AnonReviewer3 · 2020-11-01
**About representing symmetric and asymmetric functions via neural networks.  Not clear the representations are helpful or informative.  Writing too informal and rushed for me to tell**

**Rating:** 4
**Confidence:** 4

**Review:**

This paper is about representing functions $\psi : (\mathbb{R}^d)^n \rightarrow \mathbb{R}$ that are symmetric or asymmetric with respect to the permutation group $S_n$.  The aim is to consider neural networks giving only functions that symmetric or asymmetric, and to establish universality results.  The motivation comes from applications such quantum physics or computer vision with permutation symmetries.

Honestly, I am very unconvinced by this paper.  In particular, the representation for symmetric functions is essentially due to Newton ($d=1$) and to Weyl ($d > 1$).  On the other hand, the representation for asymmetric functions -- dressed up in elaborate notation and the language of Slater determinants -- seems to just be the statement that an asymmetric function is divisible by the Vandermonde determinant (when $d=1$).  Maybe there are interesting results in this paper, but the writing is so informal and apparently rushed, I simply could not tell.  Itemized comments follow.


Page 1, Footnote 4, terminology: I would suggest just sticking to one of “covariant” or “equivariant” throughout the paper

Page 1, Definition 1, notation: Although I understand the authors are using Matlab notation when they write “$\{1 : n \}$ is short for $\{1, \ldots, n\}$”, it would be more standard to abbreviate $[n] := \{1, \ldots, n\}$.

Page 2, usage of $d$, notation: Please define the meaning of $d$ before referring to this notation.

Page 3: “Functions on sets of fixed size n are equivalent to symmetric functions in n variables.”  Isn’t it symmetric functions in n variables restricted to the domain where no two variables take the same value?

Page 4: “Instead of averaging, the minimum or maximum or median or many other compositions would
also work, but the average has the advantage that smooth  lead to smooth  and  , and more
general, preserves many desirable properties such as (Lipschitz/absolute/...) continuity, (k-times)
differentiability, analyticity, etc.”  Could the authors expand on what they mean by minimum, maximum, median or other compositions to (anti)symmetrize functions?  Never heard of that.  Also, it should be remarked that the averaging operator is an orthogonal linear projection onto the subspace of (anti)symmetric functions known as the Reynolds operator in classical group invariant theory; in particular, averaging has the property that it is the identity applied to the functions that are already (anti)symmetric.

Page 4, Footnote 6: “which induces uniform convergence on compacta”. I had to google “compacta” to find out it is the plural of compact subset or compact metric space. Suffice to say this word is not widely used.

Page 4:  Do you mean $\mathcal{G}_{func} := \{g: \mathbb{R}^m \rightarrow \mathbb{R}\}$?  Also could you say more about how the basis templates $\eta_b$ are coming in?

Page 5: It is stated that computing with elementary symmetric polynomials “is numerically more stable” than with power sums.  Why?  Also why do “we need at least $m \geq n$ functional bases for a continuous representation”?

Page 5: “ Every continuous AS function can be approximated/represented by a finite/infinite linear combination of such [Slater] determinants:” Why?

Page 6, Theorem 3, theorem statement: Thus you have reduced the representation of an antisymmetric function in $n$ variables to that of $n$ symmetric functions in $n-1$ variables, at the expense of an $n \times n$ determinant.  It is not clear how computationally useful this could be, since symmetric functions in $n-1$ variables are nontrivial computationally to represent already, e.g., Theorem 2’s approach would involve $n-1$ power sums.

Page 6, Theorem 3, proof: The actual representation of $\psi({\bf{x}})$ that is constructed in the proof is almost certainly not useful, namely that it is the determinant of the matrix whose rows look like the following:

$$
\begin{pmatrix}
\psi({\bf{x}}) / \Delta({\bf{x}})  & x_1 & x_1^2 & \ldots & x_1^{n-1}
\end{pmatrix}
$$
$$
\begin{pmatrix}
\psi({\bf{x}}) / \Delta({\bf{x}})  & x_2 & x_2^2 & \ldots & x_2^{n-1}
\end{pmatrix}
$$

etc. (Sorry, can't get multi-rowed matrices to render properly in OpenReview...)

Crucial question: why is this a good way of expressing antisymmetric functions, in theory or practice?  The content here is that any asymmetric function is divisible by the Vandermonde determinant.  If the larger $d$ representation results depend on this, I don't know how much they tell me...

Page 7: “For $d > 1$, the Fermion nodes $\{{\bf X} :  \psi({\bf X}) = 0\}$ form essentially arbitrary $\psi$-dependent unions of (non-linear) manifolds partitioning $\mathbb{R}^{dn}$ into an arbitrary even number of cells of essentially arbitrary topology [Mit07].”  This language is too intuitive for me.  What does this sentence mean??

Page 7, Theorem 5, proof discussion:  “Whether this generalizes to $n > 2$  and $d > 1$  is an open problem.”  This is confusing.  Does the theorem have a restriction on $n$ and $d$?

Page 7–8, “Equivariant Neural Network” subsection: work of Risi Kondor and coauthors needs to be cited here.

Page 8, Theorem 6: It would help the reader to include in the body of the paper a more precise statement or diagram showing the construction of the approximating EMLP.

---

> ### Author Response · Authors · 2020-11-20
> **Clarification of Reviewers Questions**
>
> Thank you for your detailed review. The numbering below refers to the review’s paragraphs.
>
> 2) Newton and Weyl form the core of the proof, but the equivariance proof of EMLP is non-trivial and anything but a straightforward consequence of them. The discussion is naturally and deliberately informal but due to space restrictions dense at places, but the theorems themselves and the proofs in most parts are formal and complete, or if not, clarifications before or after the theorems are provided. The paper is neither rushed nor too informal. Regarding where the interesting results are, they are primarily concentrated in the theorems, and arguably in Table 1 in the supplementary and the discussion of open problems.
>
> 3) Except on page 1 where I discuss the various symmetry concepts more broadly, I only use ‘equivariant’ and do not mention ‘covariant’ anymore.
>
> 4) {1:n} is not taken from Matlab but may have the same origin. I find the contraction {1,…,n} → {1...n} → {1⋮n} → {1:n} more “autological” than the more fashionable [n] notation.
>
> 5) Define d=1 before usage: OK
>
> 6) In the NN literature ‘set’ is usually misused as / short-for ‘multi-set’, but that indeed should either be avoided or noted.
>
> 7) Replace the average (1/n!)∑… over permutations in (1) by min{…}, max{…}, or median{…}, e.g. ϕ(x) := min{χ(S_π(x)) : π∈S_n}, but maybe I misunderstand your question. I can mention that this is an orthogonal linear projection, called Reynolds operator. That the average (and min/max/median{χ(S_π(x)) : π∈S_n}) leave (anti)symmetric functions χ unchanged is directly implied by what I write (formally) in the line after (1) “(proof: use χ := φ or χ := ψ)”.
>
> 8) While stand-alone “compacta” is rarely used in math, the expression “uniform convergence on compacta” is a reasonably well-known notion of convergence (7000+ Google hits) corresponding to the compact-open topology.
>
> 9) Yes, G_func is the set of all functions. I am not sure what to say beyond that the basis templates η are one way to generate (multi)symmetric bases, which is particularly relevant for equivariant NN, since they “match” the local structure.
>
> 10) Afair, a representation based on elementary symmetric polynomials is less likely to require large positive and negative terms subtly canceling each other compared to using power sums as basis. The need for m≥n is explained in the provided reference [WFE+19], essentially because there are no continuous injections from ℝ^n to ℝ^m if m<n.
>
> 11) That every continuous AS function can be approximated/represented by a finite/infinite linear combination of such [Slater] determinants is a folk result in physics. In non-physics linear-algebra terms, roughly speaking, it generalizes the well-know fact that any m×m matrix can be represented as a sum of m rank-1 matrices. For AS you need rank-2 matrices, then you generalize this to tensors of order n, and then from vectors to functions. In physics jargon, this is called the Full Configuration Interaction, which can solve any n-particle Schrödinger equation exactly.
>
> 12) Reducing the problem of representing AS functions to the problem of representing symmetric/equivariant functions is as of today the _only_ feasible way of representing AS functions (for n>10 and d>1), and since the symmetric/equivariant functions can be efficiently approximated by EMLPs (the whole point of the other half of the paper) this reduction _is_ useful. There are no analogous “restricted” MLPs representing AS functions directly without the detour via determinants. There is a reason why virtually all quantum physicists use this approach, and [PSMF20] and many other papers demonstrate how/that this works in practice.
>
> 13) To prove universality, one only needs to show the existence of _some_ representation. In all likelihood the approximation found by training a NN is different from the construction used in any universality proof. The construction in Thm.3 for AS(d=1) is actually not too bad, since smooth ψ are represented by smooth (and non-singular) χ, though the loss of differentiability is disconcerting. It makes more sense to criticize the creative but unreasonable construction in the proof of Thm.5 for d>1. But as stated, in both cases it is an open problem whether this is a rectifiable artifact of the proof, or an intrinsic problem of the GSD approach itself.
>
> 14) The provided reference to slides [Mit07] contain beautiful illustrations and explanations, which are way beyond the scope of this paper.
>
> 15) Theorems 3 and 5 hold for any n. The last § of Section 5 provides a superior construction for n=2, but whether this n=2 “result” generalizes to n>2 is an open problem.
>
> 16) OK
>
> 17) Indeed, a diagram would help provided I can come up with one which is more than just showing a generic 3-Layer EMLP labeled with η, ρ, φ.

---

### Decision · Program_Chairs · 2021-01-07
**Final Decision**

**Decision:**

Reject

**Comment:**

The paper received reviews from experts in representation of invariant functions. They all have expressed concerns regarding the novelty of the technical contributions, and the lack of appropriate comparisons to existing results. This applies in particular to representation of symmetric functions using neural networks which was largely covered by previous works, as acknowledged by the authors. The authors are encouraged to consider the valuable inputs by the reviewers and revise accordingly.